# What was the impact of the first wave of COVID-19 on the delivery of care to children and adults with congenital heart disease? A qualitative study using online forums

Jo Wray ,[1,2] Christina Pagel,[3] Adrian H Chester,[4,5] Fiona Kennedy,[6] Sonya Crowe[3]

For numbered affiliations see end of article.

**Correspondence to**
Jo Wray; Jo.Wray@gosh.nhs.uk

## ABSTRACT

**Objectives** Globally, healthcare systems have been stretched to the limit by the COVID-19 pandemic. Significant changes have had to be made to the way in which non-COVID-19-related care has been delivered. Our objective was to understand, from the perspective of patients with a chronic, life-long condition (congenital heart disease, CHD) and their parents/carers, the impact of COVID-19 on the delivery of care, how changes were communicated and whether healthcare providers should do anything differently in a subsequent wave of COVID-19 infections.

**Design and setting** Qualitative study involving a series of asynchronous discussion forums set up and moderated by three patient charities via their Facebook pages.

**Participants** Patients with CHD and parents/carers of patients with CHD.

**Main outcome measures** Qualitative responses to questions posted on the discussion forums.

**Results** The forums ran over a 6-week period and involved 109 participants. Following thematic analysis, we identified three themes and 10 subthemes related to individual condition-related factors, patient-related factors and health professional/centre factors that may have influenced how patients and parents/carers experienced changes to service delivery as a result of COVID-19. Specifically, respondents reported high levels of disruption to the delivery of care, inconsistent advice and messaging and variable communication from health professionals, with examples of both excellent and very poor experiences of care reported. Uncertainty about follow-up and factors related to the complexity and stability of their condition contributed to anxiety and stress.

**Conclusions** The importance of clear, consistent communication cannot be over-estimated. Our findings, while collected in relation to patients with CHD, are not necessarily specific to this population and we believe that they reflect the experiences of many thousands of people with life-long conditions in the UK. Recommendations related to communication, service delivery and support during the pandemic may improve patients' experience of care and, potentially, their outcomes.

## Strengths and limitations of the study

► Asynchronous discussion forums enable data collection without the potential bias associated with research interviews.

► Online discussion forums facilitate those who may not be able to or want to contribute using more traditional methods of data collection to participate in research.

► During a pandemic, with limited opportunity for face-to-face contact, online discussion forums enable patients and their carers to express their views in a timely manner and offer a viable way of collecting data.

► Forum users may not be representative of the overall congenital heart disease community; they tend to be female and white.

► Patient charities moderated the forums and participants are therefore likely to be those who already engage with a patient charity.

## BACKGROUND

Since late 2019, COVID-19 has spread rapidly around the world, reaching official pandemic status in March 2020.[1] The speed with which the virus has spread and the trail of physical and psychological illness, death and economic hardship have been extensively documented in the medical and everyday press. Vast amounts of resources have been ploughed into researching the transmission, disease trajectory and risk factors associated with COVID-19. Adults with underlying health conditions have been identified as being at increased risk of developing severe and fatal disease, particularly those with pre-existing hypertension and coronary heart disease.[2] In contrast to the adult population, severe COVID-19 infection in children is rare but there is a lack of comprehensive data

on how children with underlying health conditions are affected by COVID-19.[3]

Globally, healthcare systems have been stretched to the limit and significant changes to the way in which non-COVID-19-related care has been delivered have had to be implemented. The periods of lockdown imposed in many countries and the cessation of non-essential face-to-face patient contact have necessitated rapid adjustments and adaptation to new ways of delivering and receiving care. Concerns have been raised about the impact of these changes in terms of delayed diagnosis of other health conditions,[4] delays in seeking treatment,[5] cancellations of treatment,[6] greater non-adherence to medical therapy[7] as well as increased mental health problems.[8] While health professionals and the media have been vocal about these potential consequences, far less has been heard from the patients and their families who are being directly affected.

Congenital heart disease (CHD) is one example of a chronic, life-long condition with a spectrum of severity from mild to life-threatening. It is the most common birth defect and significant improvements in diagnosis and treatment mean that currently approximately 12 million people live with CHD worldwide.[9] Both paediatric and adult patients typically require regular follow-up with specialist CHD professionals and tests of cardiac function are a cornerstone of follow-up.[10 11] But, as with other patient groups, services for patients with CHD have seen significant and abrupt changes since March 2020. In an international survey, patients with CHD and parents/carers reported significant disruption to scheduled cardiac surgery and clinic visits and high levels of psychological stress as a result of the pandemic,[12] supporting findings with other patient cohorts.[13–17] However, *how* and *who* communicates with patients with CHD and/or their carers in relation to COVID-19 has not been explored nor how patients/carers think services should be delivered in the event of a future wave of COVID-19 infection. As part of a larger study commissioned by the NHS to develop new ways of measuring the quality of CHD services for both children and adults,[18] our aim was to understand, from the perspective of patients and parents/carers, the impact of COVID-19 on the delivery of care, how changes were communicated and whether healthcare providers should do anything differently in a subsequent wave of COVID-19 infections. Our belief was that the learning and recommendations arising from this work would also be generalisable to the larger population of children and adults receiving care for other chronic health conditions.

## METHODS
### Design
A qualitative approach underpinned by an interpretivist framework was used, in which online discussion forums were employed to elicit participant (patient or parent/carer) views.

### Patient and public involvement (PPI)
A patient coresearcher (AHC) was involved with each stage of the project, including question design, data analysis and revising drafts of the manuscript. AHC also led a patient and public involvement (PPI) group set up as part of the larger overarching study (comprising three adults with CHD and one grandparent of a child with CHD), who reviewed the forum questions for content and language and the findings prior to submission. The forum questions and the presentation of the findings were revised based on feedback from the PPI group. The online discussion forums were moderated by three patient organisations, each of which contributed to the content and format of the questions. A summary of the results has been disseminated to all three charities for publication on their website and has also been disseminated to CHD services nationally via the adult CHD specialist nurse network and NHS England.

### Participants and data collection
The Children's Heart Federation, Little Hearts Matter and the Somerville Foundation, all of which are national UK charities dedicated to the support of patients with CHD and their families, facilitated and moderated one or more closed, anonymous, asynchronous online discussion groups via their Facebook pages, following an approach that we have successfully used in previous work.[19 20] We specifically chose these three charities because we wanted to collect views across age ranges (parents of younger children, teenagers and adult patients with CHD) and from those with complex and less complex CHD. Questions were developed by the authors and the content and language revised based on feedback from the charity representatives and PPI group. The charities recommended that separate forums should be facilitated for adult patients with CHD, teenage patients with CHD and parents/carers of children and young people with CHD. Each charity advertised the discussion forums on their home web page and potential participants were directed to the charity's Facebook page where they were able to access further information about the purpose of the forum, how it would be facilitated and the governance surrounding it. People interested in participating were asked to provide some basic demographic information (age, gender, ethnicity, CHD defect, location of home and specialist service, relationship to the person with CHD and age of person with CHD (for parents/carers)). Having completed this information, they were directed to the appropriate closed Facebook group, depending on participant group, where they were able to respond to the posted questions. All patients and parents/carers who wanted to participate were able to do so—there were no exclusion criteria. Participants could join (or leave) the forum at any stage and the recruitment phase lasted for the duration of the forum. The research team provided each charity with the agreed questions at the start of the process and the charity posted questions one at a time and determined when new questions should be posted

## Box 1   Questions for the adult patient forums

1. Since the start of the COVID-19 pandemic, what changes or disruptions have you experienced to your normal care for congenital heart disease?
   – Do you think these changes were appropriate in the circumstances? What did you feel about them?
   – Are you concerned about the impact of any changes on your health?
   – What did the services do well under the circumstances?
2. How were you told about the changes to services as a result of COVID-19?
   – How well were these changes communicated to you? How could this have been done better?
   – Did you have access to the information you needed? Where did you go to find out information (eg, your consultant, a charity)? How easy was it to understand the information you were given about COVID-19?
3. Looking to the future now:
   – If there is a second wave of the pandemic, should the NHS do anything differently in terms of its services for congenital heart disease compared with the first wave?
   – Which aspects of services that were disrupted are you keen to see back to normal as soon as possible?
   – Are there any changes that you would be keen to see stay even when the pandemic is over, such as telephone or online consultations?

The questions for the parent/carer and teenager forums were very similar to these, with minor wording changes to reflect those respondent groups (eg, designed to appeal to teenagers or wording appropriate for carers rather than patients).

or any prompts introduced, based on responses. When no further responses were forthcoming, the moderator posted the next question. The forums took place over a 6-week period, from August 2020 to September 2020. Questions were very similar for each participant group and each charity, with small revisions to wording to reflect the respondent group (eg, patient-relevant or carer-relevant wording). An example of the questions is provided in box 1.

### Data management and analysis

The charities removed any identifying details from the responses and provided the research team with a single transcript for each forum along with summary demographic details for each participant group. The transcripts were thematically analysed independently by four members of the research team (JW, SC, CP and AHC), following the staged approach of Braun and Clarke.[21] The first stage of familiarisation involved reading the transcripts and making initial notes, before undertaking the second stage of coding. Preliminary codes were attached to segments of data, with similar codes grouped to create themes and subthemes (stage 3) related to the perceived impact of COVID-19 on the provision of services. The research team met to discuss and review the themes and subthemes (stage 4) and to agree the descriptive names assigned to them (stage 5). The themes and suggested

recommendations were then sent with the transcripts to another member of the research team (FK) to ensure that all data related to the perceived impact of COVID-19 on the delivery of services were represented appropriately in the themes. Final revisions addressed any identified gaps or omissions.

### Ethical considerations

Each charity placed privacy notices on their websites, clarifying that participants' comments would only be visible to other members of the discussion group and the charity forum moderators and that all identifying information would be removed from discussion posts before being sent to the researchers.

## RESULTS

Five forums were run across the three charities, with 109 participants in total. One charity ran individual forums for each of the three participant groups; one charity had a single forum for adult patients; and the third charity's forum was for parents/carers of patients with CHD. Participant demographics are shown in table 1.

Three themes and 10 subthemes related to patient-related factors, individual condition-related factors and health professional/centre factors were identified, shown in figure 1 with illustrative quotes from the forums. Although there is clearly overlap between these factors, particularly in relation to communication, they represented a useful way of interpreting the data.

### Patient-related factors

For the majority of participants, routine clinics had been cancelled and appointments had been held via phone or video-link. Participants (both parents and patients) were largely accepting of these changes necessitated by the first wave of COVID-19 and considered them appropriate. They recognised that COVID-19 was new to everyone and that little was known about it initially, so they were mostly tolerant of some of the shortcomings in communication.

The theme of patient-related factors consisted of four subthemes, related to when patients were seen prior to lockdown, uncertainty about future follow-up appointments, anxiety related to any delays in treatment and their perceived safety.

#### Timing of being seen prior to lockdown

The timing of scheduled appointments was an important factor, with some patients seen just before the lockdown and highlighting that this was 'lucky'. Some patients described how they had had routine tests in the months before lockdown which reassured them when subsequent appointments were cancelled or were not face-to-face: 'My appointment was by phone rather than in person. Echo was cancelled but I'd had an MRI in February, thankfully'. In contrast, other patients who were due to be seen at around the time lockdown started decided not to attend

| Table 1 Participant characteristics | |
|---|---|
| | **Number (%)** |
| Participants: adults with CHD | 82 (75) |
| Young people with CHD | 3 (3) |
| Parents/carers of adult patients with CHD | 2 (2) |
| Parents/carers of children with CHD | 22 (20) |
| Participant gender: male | 9 (8) |
| Female | 88 (81) |
| Unknown | 12 (11) |
| Participant age group: <16 years | 1 (1) |
| 16–20 | 2 (2) |
| 21–30 | 9 (8) |
| 31–40 | 26 (24) |
| 41–50 | 28 (26) |
| 51–60 | 24 (22) |
| >61 years | 7 (6) |
| Unknown | 12 (11) |
| Age group of person with CHD: 0–1 years | 1 (1) |
| 2–5 years | 3 (3) |
| 6–10 years | 1 (1) |
| 11–15 years | 2 (2) |
| 16–18 years | 2 (2) |
| >18 years | 82 (75) |
| Unknown | 18 (17) |
| Participant ethnicity: white | 99 (91) |
| Non-white | 0 (0) |
| Unknown | 10 (9) |
| Location of specialist service: England (North East) | 3 (3) |
| England (North West) | 8 (7) |
| England (Yorkshire and the Humber) | 3 (3) |
| England (East Midlands) | 6 (6) |
| England (West Midlands) | 16 (15) |
| England (East of England) | 3 (3) |
| England (London) | 26 (24) |
| England (South East) | 6 (6) |
| England (South West) | 9 (8) |
| Wales | 1 (1) |
| Scotland | 7 (6) |
| Northern Ireland/other | 1 (1) |
| Unknown | 20 (18) |
| Location of home: England (North East) | 3 (3) |
| England (North West) | 12 (11) |
| England (Yorkshire and the Humber) | 5 (5) |
| England (East Midlands) | 5 (5) |
| England (West Midlands) | 16 (15) |
| England (East of England) | 8 (7) |

Continued

| Table 1 Continued | |
|---|---|
| | **Number (%)** |
| England (London) | 8 (7) |
| England (South East) | 13 (12) |
| England (South West) | 16 (15) |
| Wales | 4 (4) |
| Scotland | 8 (7) |
| Northern Ireland/other | 1 (1) |
| Unknown | 10 (9) |
| Complexity of CHD: single ventricle condition | 21 (19) |
| Biventricular condition | 83 (76) |
| Unknown | 5 (5) |

A number of participants chose not to provide some or any demographic information.

and cancelled their appointments, preferring instead to wait.

### Uncertainty about future follow-up

Participants expressed uncertainty about when they would be seen and this was exacerbated if communication from their specialist centre was poor. Reported concern and/or distress were notable in patients who were newly diagnosed or who were in the process of transferring between centres: 'As I was moving from one hospital to another I had nothing (information) as neither hospital took responsibility for me'. Many people described the challenges of getting information about follow-up arrangements, illustrated by one patient: 'I spent many months going round in circles and being passed from pillar to post'.

### Anxiety about delay in treatment or in diagnosing deterioration

Linked to uncertainty about future follow-up was the subtheme of anxiety related to delays and the consequences of these. Participants described feeling anxious and stressed about delays in treatment, diagnosis or identifying any deterioration in their condition. Prior to lockdown a number of patients were waiting for treatment or had planned surgery for later in the year and this was a significant concern: 'The next surgery was 'urgent' and was scheduled but then cancelled due to COVID… my delayed treatment through COVID has been a huge disappointment, cause of stress and who knows what consequences the wait has had'. Some parents talked about the responsibility of monitoring their child for signs of deterioration or the onset of problems and having to decide when their child should be seen: 'My daughter is currently in between operations and it's worrying…they told us to look out for signs such as low sats, energy levels and weight. I just feel it's pressure on me to judge when she will next be seen. I am also worried…it's [COVID] going to delay future surgeries, cath labs and MRI'. For some, this stress was intensified by the loneliness brought

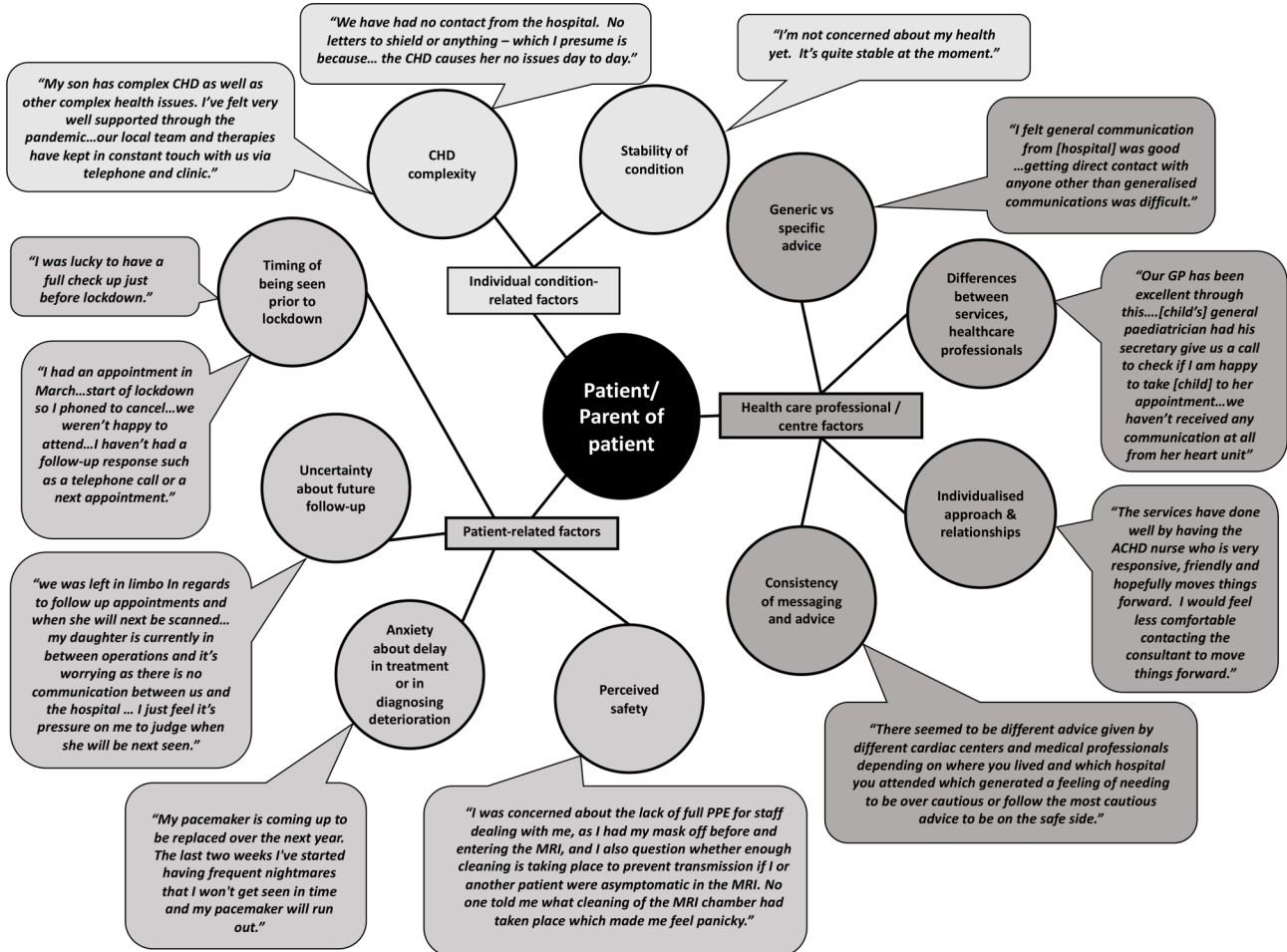

**Figure 1** Factors influencing patients'/parents' experiences of the impact of COVID-19 on service delivery and care.

about by the enforced isolation: 'I've yet to see anyone. I got a shielding letter, that was it. I've found life very lonely and frightening'. A number of patients also described feeling that, as non-COVID-19 patients, they were not a priority: 'I felt I was being ignored and that unless you were a person with COVID no-one wanted to know', with potential consequences for their ongoing care and health.

### Perceived safety

A number of participants talked about safety, both in terms of perceived risks to their health from being in the hospital environment or using public transport as well as the risks of not being seen face-to-face and getting the necessary tests and/or interventions: 'COVID stopped me going to (hospital) for my consultation. This has its plus points and minus points. The changes under the circumstances were fine because I would have had to have travelled on public transport and it's something that I wasn't willing to do. However, I prefer to go to the hospital as it puts my mind at ease when they can do the necessary tests required'. The need for balance was summed up by one participant: 'Things have to continue but in the safest way possible for all'.

### Individual condition-related factors

As with many other chronic health conditions, there is a spectrum of both complexity and stability of CHD and these two factors seem to be important determinants of how COVID-19 was perceived to impact patients.

### CHD complexity

A number of those with more complex CHD were very well supported by their specialist service as well as local primary and secondary care services. They described receiving regular phone calls and written information and, where necessary, individual arrangements for tests at local surgeries or hospitals: 'The practice nurse called every couple of weeks to check we had all we needed... (child) normally has blood tests every 3 months at the local hospital but this couldn't happen so the GP arranged for it to be done at the surgery with one of their nurses...and arranged a time when the surgery was empty.' In contrast, some others, particularly those with less complex CHD, reported having no contact from their specialist centre and frequently felt that they had to chase for information about changes to services and guidance about shielding: 'I had nothing from anyone. I just found out on my own by looking on the BBC website mainly. I wasn't informed

about changes and had to phone (specialist centre) to find out'.

### Stability of condition

Although they wanted information about arrangements, those patients whose conditions were stable generally expressed low levels of concern about their health and the impact on it of any changes to their care: 'My condition is stable and I am well. So for me, COVID-19 hasn't had any impact in terms of cardiac care'. For patients who were unstable or who had developed new symptoms, however, the added uncertainty about how and when they might be seen was particularly stressful: 'It's horrible knowing I have a critical illness and knowing I need surgery but not knowing how bad it is. For 5 months now I've been in limbo and frightened'.

### Healthcare professional/centre factors

The theme of healthcare professional and centre factors comprised four subthemes related to consistency of messaging and advice, generic versus specific advice, differences between services and individualised approaches and relationships.

### Consistency of messaging and advice

Communication was the factor that evidently had the biggest impact on patients and parents and how they perceived COVID-19 to have affected them or their child. There was general consensus that messaging and advice had been inconsistent, with different centres and different professionals offering different advice about the same thing: 'Communication from centres about shielding was very contradictory' and 'Hospitals were telling people different things. Some hospitals said single ventricle had to shield whereas others said they didn't have to'. One parent described the advice she was given about her child: 'Contacted GP to ask if he should shield and was told no…(then) told he should be shielding…' Some patients also received letters from the government identifying them as extremely vulnerable and that they should be shielding which directly contradicted the advice given to them by their specialist team, whereas others were told by their specialist team that they should shield but did not receive any information to that effect from the government. A number of participants commented on the vital role played by charities in providing information and support to patients and their families, although this also highlighted differences between specialist centres in the guidance they were providing: 'There were lots of people getting distressed…because they had heard nothing from their centre when other people had received guidance letters…more consistency in level of service would be useful'.

### Generic versus specific advice

Participants made a distinction between general advice and patient or condition specific advice, the latter of which was generally more difficult to access: 'I got a standard email about COVID-19 and my risk. Later on I got personal advice from my consultant and the nurses'. Another patient described how they were initially notified by their hospital about COVID-19 generally and that was followed up by a call from the CHD nurse to make sure they understood and were happy with what they had to do.

### Differences between services and healthcare professionals

Participants described variation in the contact they had had with different professionals involved in their care: some described the excellent support they had received from their GP but a complete lack of communication from the specialist centre, whereas for others the reverse was true and it was the cardiac team who were supportive in the absence of any contact from their GPs or local teams. As one participant commented, 'not all medically trained individuals are reading off the same hymn sheet'.

### Individualised approach and relationships

Some respondents reported that clinicians, particularly cardiac specialist nurses (CLNs) who knew them/their medical history, were proactive and responsive to their queries and this was valued by patients and parents: 'I have no concerns as I find the CLNs are accessible by phone or email and I'm confident that if I had any issues I would be seen sooner'. Another described how 'their' nurse had been really helpful with advice about COVID-19, highlighting the value of being able to contact professionals who knew them.

### What should healthcare providers do differently in a subsequent wave of COVID-19?

While there was a degree of acceptance and understanding about changes to services during the first wave of COVID-19, participants expressed very different expectations for managing the on-going situation and clearly articulated that, as awareness and knowledge about COVID-19 are increasing all the time, they are likely to be far less understanding and tolerant of poor communication, delays and cancellations. A number of participants expressed concerns about the big backlog of appointments and the likelihood that quite a few patients will have deteriorated, resulting in additional health issues for them and additional input and costs incurred by the NHS: 'I understand it must be very difficult but if we have a second wave I think appointments for those awaiting surgery should go ahead. I understand it's dangerous, however leaving symptomatic patients without an appointment could be catastrophic. And would subsequently put more pressure/expense on the NHS.'

Based on participants' experiences and responses about what healthcare professionals should do differently in any subsequent wave of COVID-19, a series of recommendations has been developed in relation to four domains: generic communication, patient-specific communication, service delivery and support (box 2).

### DISCUSSION

The impact of COVID-19 on the delivery of services to patients with CHD in the UK has been significant, with

---

**Box 2  Recommendations for improving patients' experience of care and, potentially, their outcomes, based on what participants told us in the discussion forums**

**Communication—generic**
► Consistent information from all healthcare providers in relation to condition-specific advice.
  – Includes all hospitals, GPs, community services, etc.
  – Should be routinely provided to patients with a particular condition, wherever they receive their care.
► Produce and share information about the latest guidance and recommendations with those around the patient.
  – Includes, but not limited to, schools, nurseries and employers.
  – Ensure guidance is condition-specific and accessible to patients, to facilitate sharing.

**Communication—patient specific**
► Clear advice and guidance about shielding (personalised to individual rather than generic).
  – Provided to all patients via a range of media (email, letter, easy read, text message±telephone).
► Proactive communication with patients via email or telephone.
  – To check in with them.
  – To update them about any changes.
  – Determined by individual patient circumstances and need.
► Dedicated email address/phone line with answerphone for patients to call with concerns or questions.
  – Checked and responded to regularly by someone familiar with their individual case.
  – Provides clear information about how frequently messages are checked and when a response can be expected.

**Service delivery**
► Regular updates about services.
  – Any curtailment of services, estimated delay times, safety precautions being put in place.
► Greater flexibility for tests being done locally, more remote monitoring.
► Telehealth for some/quick catch-ups or where face to face is not necessary.
  – For communication of routine test results.
  – Intermediate appointments for patients seen very frequently.
  – Benefits in terms of reducing travel, time efficiency and safety.
► Face to face where indicated/necessary.
  – For medical tests.
  – Where patients have complex needs.
  – Underpinned by patient choice about how and where their care should be delivered.
► Protection of specialist services, COVID-19 free beds.
► Individualised approach to patient care and follow-up.
  – Tailored to diagnosis.
  – Dependent on where an individual is in terms of their care pathway—for example, waiting for a treatment intervention versus requiring routine check-up.

**Support**
► Increased access to online support.
  – Signposting to existing support groups and websites.
► Provision of access to.
  – Support meetings.
  – Videos made by health professionals.
  – Other resources established in response to COVID-19.

Continued

**Box 2  Continued**

Although generated from research related to congenital heart disease, we believe that these recommendations are relevant for patients with any underlying health conditions.

---

consequences for both patients and their parents/carers in terms of anxiety and stress. Our findings support those of Cousino and colleagues,[12] who also identified high levels of disruption to routine CHD services and resulting effects on mental health, although, in contrast to this latter study, we did not find a high level of concern expressed about returning to face-to-face appointments. On the contrary, many respondents in our study wanted face-to-face appointments to be reinstated. Although some patients were concerned about their safety in the hospital environment because of the risks associated with COVID-19, as has been reported by parents of children with cancer[22] and asthma,[23] fear about getting COVID-19 was not a dominant theme in our study. Of note, however, is that other studies explicitly asked respondents about their anxiety related to getting COVID-19 and we did not do this.

We were also interested in *how* patients found out about changes to their care and the importance of clear, consistent communication cannot be over-estimated. Lack of consistency in guidance, confused and contradictory messaging and uncertainty characterised many responses, mirroring the national picture in relation to communication about COVID-19[24] as well as results from studies with other patient groups.[25] A number of patients were surprised that they had not had any contact from their specialist centre, particularly those with more complex CHD who are typically relatively high users of healthcare, indicating that their expectations about communication with their specialist team were not met, and this mismatch between expectations and reality is likely to have contributed to higher stress levels.[26] The findings from this study suggest a somewhat mixed picture: some respondents reported being very satisfied with arrangements and described excellent communication and care; others reported some positive aspects of care delivery but they also expressed examples where communication, particularly, had been poor or inconsistent; a third group were very dissatisfied and disappointed with the lack of communication and disruption to their care. Participants also described examples of good practice, such as the responsiveness of the clinical nurse specialists, the online support groups facilitated by psychologists and other health professionals and the freely available YouTube educational videos developed by their consultants. One contributory factor to the different patterns of communication may have been regional levels of COVID-19 infection, with those centres in areas with high levels of infection potentially finding it harder to keep up with communication, particularly if staff were redeployed to

provide front-line care in other areas or were working remotely.

## Limitations

Facebook has been used in a variety of ways in numerous studies and remains a dominant player in the social media milieu.[27] Although we specifically chose a method of data collection to increase the accessibility of the research to potential participants and did achieve good diversity in terms of where participants lived and their specialist centre, participants did not reflect a broad range of ethnic groups or gender. This may be of particular salience in light of the growing body of evidence that people from black Asian and minority ethnic (BAME) groups have been disproportionately affected by COVID-19, including experiencing higher rates of mortality due to COVID-19[28] and heightened levels of anxiety.[29] Even if this is not shown to be the case for patients with CHD, such knowledge is likely to contribute to higher levels of anxiety in BAME individuals and may drive greater social isolation and disengagement with healthcare, which is an important consideration for specialist centres and the wider health service. The lack of participation from BAME groups reflects a recognised problem that they are less likely to engage with, and participate in, research than their white British counterparts[30] and speaks to the need for targeted strategies to involve, recruit and retain BAME individuals in research projects.

Charities (not limited to those who moderated the discussion forums in this research) were identified as having a vital role in providing support and information to patients and families during the first wave of COVID-19 and at times were the *only* perceived source of information and support. This also highlights a bigger issue of inequity as it will only be those patients and families who are willing and able (through familiarity and adequate language and literacy skills as well as internet resources) to access charity resources who will be able to benefit from them. Furthermore, many of those who are excluded from this will also be those who are less well informed and have less awareness of guidance about issues related to COVID-19. In light of the important role that they play, it may also be timely for charities to reflect on how to increase their appeal to, and membership from, BAME and other under-represented communities.

Our findings, while collected in relation to patients with CHD and their parents/carers, are not necessarily specific to this population and we believe reflect the experiences of many thousands of people with life-long conditions in the UK. Healthcare delivery changed significantly during lockdown and beyond, and as with all changes, there are lessons to be learnt. The recommendations that have been developed from what participants told us in the discussion forums, would, we think, improve patients' experience of care and, potentially, their outcomes. Monitoring of experiences and outcomes should be routinely undertaken, particularly at a time when patients are more vulnerable, to evaluate the impact of changes to service delivery and support as well as the implications for resource utilisation and to enable further changes to be responsive to patient need. A key element of the recommendations is flexibility and individualisation and our findings clearly demonstrate the diversity in responses to COVID-19, at both a patient and institutional level. We believe the proposed recommendations, monitoring and evaluation are applicable to any patients with underlying health conditions and some, particularly those related to communication, would likely reap large benefits for relatively little input. While the data were collected specifically in relation to COVID-19 and the learning has come from patients' experiences of care during the lockdown, a number of these recommendations are relevant to the wider delivery of care to patients with chronic underlying health conditions and reflect principles of good communication and service delivery.

**Author affiliations**
[1]Heart and Lung Directorate, Great Ormond Street Hospital for Children NHS Foundation Trust, London, UK
[2]Research Department of Children's Cardiovascular Diseases, Institute of Cardiovascular Science, University College London, London, UK
[3]Clinical Operational Research Unit, Department of Mathematics, University College London, London, UK
[4]Heart Valve Research Group, The Magdi Yacoub Institute, Heart Science Centre, Harefield, UK
[5]Myocardial Function, National Heart and Lung Institute, Imperial College London, London, UK
[6]Adult Congenital Heart Disease Department, Barts Health NHS Trust, London, UK

**Acknowledgements** The authors thank The Somerville Foundation, Little Hearts Matter and the Children's Heart Federation for recruiting participants and moderating the forums, and for their contributions, together with the PPI group, to the development and review of the forum questions.

**Contributors** JW and SC contributed to the design of the study and undertook the initial analysis of the transcripts; wrote the initial draft of the manuscript and approved the final version. CP and AHC contributed to the design of the study and undertook the initial analysis of the transcripts; revised the manuscript and approved the final version. FK contributed to the design of the study; independently checked that all data related to the perceived impact of COVID-19 on the delivery of services were represented appropriately in the themes; revised the manuscript and approved the final version.

**Funding** JW is supported by the Great Ormond Street Hospital NIHR Biomedical Research Centre. This study is independent research funded by the National Institute for Health Research (Policy Research Programme, Congenital Heart Audit: Measuring Progress In Outcomes Nationally (CHAMPION), PR-R20-0318-23001).

**Disclaimer** The views expressed in this publication are those of the author(s) and not necessarily those of the NHS, the National Institute for Health Research or the Department of Health and Social Care.

**Competing interests** None declared.

**Patient and public involvement** Patients and/or the public were involved in the design, or conduct, or reporting or dissemination plans of this research. Refer to the Methods section for further details.

**Patient consent for publication** Not required.

**Ethics approval** The Research Ethics Committee confirmed that ethical approval was not required because the forums were managed by the charities.

**Provenance and peer review** Not commissioned; externally peer reviewed.

**Data availability statement** No data are available. No additional data are available due to the potential for identification of participants from their qualitative comments.

**ORCID iD**
Jo Wray http://orcid.org/0000-0002-4769-1211

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
