## [Reviewer comments · BMJ Open]

ARTICLE DETAILS

TITLE (PROVISIONAL)	What was the impact of the first wave of COVID-19 on the delivery of care to children and adults with congenital heart disease? A qualitative study using online forums.
AUTHORS	Wray, Jo; Pagel, Christina; Chester, Adrian; Kennedy, Fiona; Crowe, Sonya

VERSION 1 – REVIEW

REVIEWER	Willems, Ruben University of Ghent, Public health and primary care
REVIEW RETURNED	10-Mar-2021

GENERAL COMMENTS	The authors present a qualitative study on the impact COVID-19 has on the treatment of CHD patients. They have made use of a series of discussion forums which were moderated by three patient charities via their Facebook pages: an interesting, creative solution in COVID-19 times which can be a complementary methodology next to other qualitative methodologies as the patient selection bias is differently. Another strength of the study is the creative way of presenting the (sub)themes and related quotes in a Figure, very well done. I have two main issues with the paper which should be addressed before it can be accepted for publication: · The discussion section does not follow the high quality of the introduction, methods, and results section. I believe that the results should be placed in perspective to other literature. This is completely lacking (there are two references included in the discussion section, both related to a methodological issue, and zero content-related). Specifically:- The authors stated that the data were collected in relation to COVID-19 but that it might be applicable to the wider delivery of care. Please relate this to other qualitative papers on the CHD-population. Did you find similar or opposite results...?- The data were collected in CHD patients but the authors stated that the results might be of importance for other chronic patient populations as well. Aren't there any other papers published yet about the impact of COVID-19 on chronic patient populations?- The data were collected in Summer 2020, before the 2nd COVID-19 wave raged in the UK (and in Europe in general). Is it
--

	possible to reflect on how care was organized in this second wave: was it for instance more focused on non-COVID-19-patients? So in general, was the approach more conform the recommendations as stated in the paper? · The abstract should be rewritten. It does not present any results in its current form. What are the main results and what are the conclusions to be drawn? Other (minor) remarks/suggestions: · P7, line 34-39: the charity received the questions from the researchers and questions were released one by one (I suppose?), but it's not completely clear to me when the next question was released (after x answers on a question?). So it's possible that some people answered the first but not the last questions? · P9, line 34: sample size of 109 <-> 111 in abstract. · P9, line 34: three charities, five forums. Why did some charities start multiple forums? (something for the methods section?) · Results section: 'some', 'many', 'a number of', 'others',... try to be consistent. In qualitative research, there is the ongoing debate if researchers should count. It is my personal opinion that it is worthwhile to indicate how many patients/interviewees elaborated on a specific theme. It makes qualitative research more transparent (e.g., PMID: 31489832). But as said, it's up for debate. Anyway, I would recommend to use the same wording consistently throughout the results section. · Discussion section: there are some quotes inserted in the discussion section. I understand it fits in nicely but it's my opinion that quotes are results, and thus belong in results section, not in the discussion section.
--	--

REVIEWER	Bay, Annika Umeå Universitet Medicinska fakulteten, Nursing
REVIEW RETURNED	11-Mar-2021

GENERAL COMMENTS	The impact of the first wave of Covid-19 on those with lifelong conditions: a case study of congenital heart disease Overall the study presents very interesting and important findings. The Covid-19 pandemic will have many consequences regarding follow-ups, and this case study fills a gap concerning lack of follow-ups. The paper is well written and nice to read. The method section and discussion sections needs extra attention, I lack a more detailed description overview from the analysis procedure. Although I like the Figure, I miss the detailed text that explains how you concluded the findings. There are very few references, and a lot of them are from your own earlier research. Please include other research as well, it will strengthen your article. Abstract Design and setting: Should state what type of study it is, please clarify. Background Page 5, Line 50; Maybe some more information about CHD, and please have som references. I miss the aim of the study in the end of background.
--

	Methods The method section needs to be reviewed, please clarify how the analysis was performed. Page 7, line 3; You are referring to work who has been made by your self, do you have any other references who says that it is a successfully way to use discussions groups on Facebook, has it been made by others? Page 7, line 57; What kind of qualitative method did you use? Please explain and refer to a appropriate method. Results Concerning the results, I find it interesting and important, your themes are very close to the text. Maybe it has to be elaborated more? In the method section, page 8, line 1; you are writing that you created sub-themes but you do not really mention those in the results, please elaborate. I also would have thought it interesting follow the analysis in a table, describing the process with data extract and codes. Discussion In this section, I miss discussion towards other authors, although the Covid-19 situation is new and there are few articles publish in that area, but I think that your findings can be put in a different context? There are studies done on lack of follow-ups, and much is done on transitions from child to adult care. Please discuss your findings against other authors. Figure Please name the figure (The text that you have under Figure legend).
--	--

REVIEWER	Sood, Erica Alfred I DuPont Hospital for Children, Division of Behavioral Health
REVIEW RETURNED	20-Mar-2021

GENERAL COMMENTS	This is a timely study that has several strengths including involvement of patient/family stakeholders in study design, implementation, and interpretation. Limited sample diversity and the likelihood that the study sample is not representative of the broader CHD population due to the recruitment strategy are major limitations but are discussed appropriately in the Discussion section of the paper. Tables and figures are appropriate and helpful, but the Results and Discussion sections would benefit from revision, as described below. Introduction: -I would recommend changing "over the last 9 months" to "since the start of the COVID-19 pandemic" or "since March 2020" as the 9 month estimate will not be accurate at the time of publication. -The introduction would benefit from review of other published studies focused on patient and caregiver perspectives of
---

healthcare delivery during the COVID-19 pandemic. For example, Cousino et al. published in *Cardiology in the Young* in December 2020, which is also focused on CHD. Similar studies with other chronic illness populations could also be mentioned. Then propose what gaps the current study fills in the literature.

Methods:

-In the PPI section, the phrase "who reviewed the forum questions and findings" is not clear. Did they simply review the questions and findings or were they involved in the process of creating the questions and analyzing the qualitative data? More information is presented in the following section, but it would be helpful to revise this sentence to provide a bit more detail here as well.

-Were all interested patients/caregivers admitted to the private online forum and included in analyses? Was anyone determined to be ineligible or excluded? Was there a specific period of enrollment or could patients/caregivers join at any time even once questions were being posted? Were there any efforts made to increase the diversity and representativeness of the sample?

Results:

-The Results section states "Five forums were run across the three charities, with 109 participants in total." However, the abstract reports 111 participants total. Also, the methods section states: "The charities recommended that separate forums should be facilitated for adult patients with CHD, teenage patients with CHD and parents/carers of children and young people with CHD" suggesting three groups, not five. Please clarify the composition of the five groups. How many adult patient, teen patient, and parent/caregiver groups across how many charities?

-I found it very difficult to make sense of the results based on the text alone. It is not clear what the subthemes are or how the results fit into the three larger themes without understanding of the subthemes. The figure is much more clear, but readers should be able to understand the results by reading the text rather than having to rely on the figure to understand (although the figure can certainly complement what is written in the text). I would recommend a major revision of the Results section to clearly define the subthemes and ideally provide some text regarding each subtheme. This will lengthen the Results section, but will allow readers to understand the Results by reading the text, which seems important.

-Similarly, one of the aims is to understand patient/parent perspectives regarding "what should happen in any subsequent wave of COVID-19." However, this does not seem to be addressed in the Results section. Although the Table with recommendations is helpful, the text should also address this.

Discussion:

-The Discussion is very brief and seems to primarily repeat results and even includes additional quotes. Instead it would be helpful to include a discussion of how these findings relate to other relevant literature (such as those presented in the Introduction - see suggestions above), and well as clinical and research implications.

Reviewer: 1

Dr. Ruben Willems, University of Ghent

The authors present a qualitative study on the impact COVID-19 has on the treatment of CHD patients. They have made use of a series of discussion forums which were moderated by three patient charities via their Facebook pages: an interesting, creative solution in COVID-19 times which can be a complementary methodology next to other qualitative methodologies as the patient selection bias is differently. Another strength of the study is the creative way of presenting the (sub)themes and related quotes in a Figure, very well done.

Thank you for your positive feedback.

I have two main issues with the paper which should be addressed before it can be accepted for publication:

- The discussion section does not follow the high quality of the introduction, methods, and results section. I believe that the results should be placed in perspective to other literature. This is completely lacking (there are two references included in the discussion section, both related to a methodological issue, and zero content-related). Specifically:

- The authors stated that the data were collected in relation to COVID-19 but that it might be applicable to the wider delivery of care. Please relate this to other qualitative papers on the CHD-population. Did you find similar or opposite results...?

There are still limited data reporting the patient perspective in relation to COVID-19 and the delivery of care to CHD patients but as suggested we have placed our findings in the context of the work that has been published.

- The data were collected in CHD patients but the authors stated that the results might be of importance for other chronic patient populations as well. Aren't there any other papers published yet about the impact of COVID-19 on chronic patient populations?

We have included further reference to other published work in other disease groups, as suggested.

- The data were collected in Summer 2020, before the 2nd COVID-19 wave raged in the UK (and in Europe in general). Is it possible to reflect on how care was organized in this second wave: was it for instance more focused on non-COVID-19-patients? So in general, was the approach more conform the recommendations as stated in the paper?

We agree with Dr Willems that it would be interesting to reflect on how care was organised in the second wave of the pandemic, since the data in our study were collected. Unfortunately it is beyond the scope of this paper to accurately reflect how care was organised in the UK, given the number of centres and the different approaches to delivering care already identified, and we would only have limited, anecdotal evidence at best at this stage.

- The abstract should be rewritten. It does not present any results in its current form. What are the main results and what are the conclusions to be drawn?

We have rewritten the abstract as suggested to include the main results and conclusion.

Other (minor) remarks/suggestions:

- P7, line 34-39: the charity received the questions from the researchers and questions were released one by one (I suppose?), but it's not completely clear to me when the next question was released (after x answers on a question?). So it's possible that some people answered the first but not the last questions? We have added further detail to the methods to improve the clarity of this section. Dr Willems is correct in that questions were posted one by one and a new question was posted when no further responses were

forthcoming. As the reviewer states, some respondents could (and did) answer the first questions but not the later questions.

- P9, line 34: sample size of 109 <-> 111 in abstract.

Thank you – we have corrected the typographical error in the abstract.

- P9, line 34: three charities, five forums. Why did some charities start multiple forums? (something for the methods section?)

We have added further detail to clarify that one charity ran three forums (one for each participant group) and the other two charities ran one forum each.

- Results section: 'some', 'many', 'a number of', 'others',... try to be consistent. In qualitative research, there is the ongoing debate if researchers should count. It is my personal opinion that it is worthwhile to indicate how many patients/interviewees elaborated on a specific theme. It makes qualitative research more transparent (e.g., PMID: 31489832). But as said, it's up for debate. Anyway, I would recommend to use the same wording consistently throughout the results section.

We did not set out to undertake a content analysis and provide numbers of respondents who reported on a specific theme and so we have not included this information. We have rewritten the results at the suggestion of both the other reviewers and whilst we appreciate the point that Dr Willems makes in relation to consistency of terminology, we think that to use just one determiner detrimentally impacts the flow and readability and so we have not done this entirely. We are also not aware that consistency in determiner terminology is particularly recommended in the reporting of qualitative studies.

Discussion section: there are some quotes inserted in the discussion section. I understand it fits in nicely but it's my opinion that quotes are results, and thus belong in results section, not in the discussion section. Thank you – we agree that quotes should be in the results section and have removed them from the discussion accordingly.

Reviewer: 2

Dr. Annika Bay, Umeå Universitet Medicinska fakulteten

Overall the study presents very interesting and important findings. The Covid-19 pandemic will have many consequences regarding follow-ups, and this case study fills a gap concerning lack of follow-ups. The paper is well written and nice to read.

Thank you for this feedback.

The method section and discussion sections needs extra attention, I lack a more detailed description overview from the analysis procedure. Although I like the Figure, I miss the detailed text that explains how you concluded the findings.

There are very few references, and a lot of them are from your own earlier research. Please include other research as well, it will strengthen your article.

Thank you for these suggestions. We have added extra detail to the methods section, rewritten the results and added to the discussion to include additional references to other research.

Abstract

Design and setting: Should state what type of study it is, please clarify.

We have added in the study type.

Background

Page 5, Line 50; Maybe some more information about CHD, and please have some references. I miss the aim of the study in the end of background.

We have included some further references about CHD and made the study aim more explicit.

Methods

The method section needs to be reviewed, please clarify how the analysis was performed.

We have added further detail about the analysis.

Page 7, line 3; You are referring to work who has been made by your self, do you have any other references who says that it is a successfully way to use discussions groups on Facebook, has it been made by others?

We are not aware of other studies in which Facebook has been used in the same way that we used it but we have included a further reference to the use of Facebook in research in the discussion.

Page 7, line 57; What kind of qualitative method did you use? Please explain and refer to a appropriate method.

We have added further detail about the thematic analysis method that we used, following the approach of Braun and Clarke.

Results

Concerning the results, I find it interesting and important, your themes are very close to the text. Maybe it has to be elaborated more?

In the method section, page 8, line 1; you are writing that you created sub-themes but you do not really mention those in the results, please elaborate.

I also would have thought it interesting follow the analysis in a table, describing the process with data extract and codes.

We have rewritten the results section and provided more detail and clarification about the subthemes with accompanying quotes to make the analysis process clearer.

Discussion

In this section, I miss discussion towards other authors, although the Covid-19 situation is new and there are few articles publish in that area, but I think that your findings can be put in a different context? There are studies done on lack of follow-ups, and much is done on transitions from child to adult care. Please discuss your findings against other authors.

Thank you – we have added further detail to the discussion with additional references to other published literature. We have not specifically added references about transition from paediatric to adult care as that was not a focus of this work and it was not discussed by our forum participants in the context of delivery of care but we have added in further references to the importance of follow-up for patients with CHD.

Figure

Please name the figure (The text that you have under Figure legend).

We have added the figure legend details to the actual figure.

Reviewer: 3

Dr. Erica Sood, Alfred I DuPont Hospital for Children

Introduction:

-I would recommend changing "over the last 9 months" to "since the start of the COVID-19 pandemic" or "since March 2020" as the 9 month estimate will not be accurate at the time of publication.

Thank you – we have amended the text as suggested.

-The introduction would benefit from review of other published studies focused on patient and caregiver perspectives of healthcare delivery during the COVID-19 pandemic. For example, Cousino et al. published in *Cardiology in the Young* in December 2020, which is also focused on CHD. Similar studies with other chronic illness populations could also be mentioned. Then propose what gaps the current study fills in the literature.

Thank you – we have added some additional references to published work, including the work of Cousino and colleagues and findings about other chronic illness populations.

Methods:

-In the PPI section, the phrase "who reviewed the forum questions and findings" is not clear. Did they simply review the questions and findings or were they involved in the process of creating the questions and analyzing the qualitative data?

More information is presented in the following section, but it would be helpful to revise this sentence to provide a bit more detail here as well.

We have added further detail as suggested.

-Were all interested patients/caregivers admitted to the private online forum and included in analyses? Was anyone determined to be ineligible or excluded? Was there a specific period of enrollment or could patients/caregivers join at any time even once questions were being posted? Were there any efforts made to increase the diversity and representativeness of the sample?

Thank you for these suggested additions. We have now included information about inclusion criteria and how and when participants could join or leave. We acknowledge the lack of diversity of our sample but given the approach we used (charity-moderated online discussion forums) we could only try and increase the diversity and representativeness through the charity websites, which had obvious limitations which we have outlined in the discussion.

Results:

-The Results section states "Five forums were run across the three charities, with 109 participants in total." However, the abstract reports 111 participants total. Also, the methods section states: "The charities recommended that separate forums should be facilitated for adult patients with CHD, teenage patients with CHD and parents/carers of children and young people with CHD" suggesting three groups, not five. Please clarify the composition of the five groups. How many adult patient, teen patient, and parent/caregiver groups across how many charities?

-I found it very difficult to make sense of the results based on the text alone. It is not clear what the subthemes are or how the results fit into the three larger themes without understanding of the subthemes. The figure is much more clear, but readers should be able to understand the results by reading the text rather than having to rely on the figure to understand (although the figure can certainly complement what is written in the text). I would recommend a major revision of the Results section to clearly define the subthemes and ideally provide some text regarding each subtheme. This will lengthen the Results section, but will allow readers to understand the Results by reading the text, which seems important.

-Similarly, one of the aims is to understand patient/parent perspectives regarding "what should happen in any subsequent wave of COVID-19." However, this does not seem to be addressed in the Results section. Although the Table with recommendations is helpful, the text should also address this.

Thank you. We have rewritten the results section as suggested to provide clarity about the individual five forums as well as defining the subthemes with accompanying quotes. We have now mentioned the recommendations in the results section and the four themes which underpin them.

Discussion:

-The Discussion is very brief and seems to primarily repeat results and even includes additional quotes. Instead it would be helpful to include a discussion of how these findings relate to other relevant literature (such as those presented in the Introduction - see suggestions above), and well as clinical and research implications.

We have rewritten the discussion to remove repetition of the results and the additional quotes and to include further reference to other relevant literature, as suggested. We have also added in further detail about the need for ongoing evaluation of outcomes and experiences.

In view of the extensive additions requested, particularly to the results section but also to the methods and discussion, we have now exceeded the suggested word limit of 4000 words and our manuscript is currently at 4255 words. We hope that this is acceptable and note that Dr Sood identified that her suggested revisions would lengthen the results section but that it was important for readers to be able to understand the text by reading the text only, rather than relying on the figure.

VERSION 2 – REVIEW

REVIEWER	Willems, Ruben University of Ghent, Public health and primary care
REVIEW RETURNED	22-Jun-2021

GENERAL COMMENTS	Thank you for addressing my comments. The discussion section is still rather short to my opinion, but definitely of good quality and surely acceptable for publication.
---

REVIEWER	Bay, Annika Umeå Universitet Medicinska fakulteten, Nursing
REVIEW RETURNED	17-Jun-2021

GENERAL COMMENTS	The manuscript has improved after the revision and I now have nothing that I think should be improved.
--

REVIEWER	Sood, Erica Alfred I DuPont Hospital for Children, Division of Behavioral Health
REVIEW RETURNED	14-Jun-2021

GENERAL COMMENTS	This revised manuscript is much improved and all of my prior comments have been addressed. My only remaining concern involves this sentence in the Introduction: "However, how and who communicates with patients with CHD and/or their carers in relation to COVID-19 has not been explored nor how patients/carers think services should be delivered in the event of a future wave of COVID-19 infection." The phrase "how and who communicates" does not seem grammatically correct and should be revised prior to publication.
---